# Separating selection from mutation in antibody language models

**Frederick A Matsen IV[1,2,3,4]\*, Will Dumm[1], Kevin Sung[1], Mackenzie M Johnson[1], David H Rich[1], Tyler N Starr[5], Yun S Song[6], Julia Fukuyama[7], Hugh K Haddox[1]**

[1]Computational Biology Program, Fred Hutchinson Cancer Center, Seattle, United States; [2]Department of Genome Sciences, University of Washington, Seattle, United States; [3]Department of Statistics, University of Washington, Seattle, United States; [4]Howard Hughes Medical Institute, Seattle, United States; [5]Department of Biochemistry, University of Utah School of Medicine, Salt Lake City, United States; [6]Computer Science Division and Department of Statistics, University of California, Berkeley, Berkeley, United States; [7]Department of Statistics, Indiana University, Bloomington, United States

## eLife Assessment

This **fundamental** study introduces a new biology-informed strategy for deep learning models aiming to predict mutational effects in antibody sequences. It provides **convincing** evidence that separating selection from the nucleotide-level mutation process improves performance over the objectives of protein language models inspired by natural language processing. This paper should be of interest to computational immunologists, but also to the broader community interested in deep learning for biological sequence data and evolution.

**\*For correspondence:** matsen@fredhutch.org

**Competing interest:** The authors declare that no competing interests exist.

**Preprint posted** 22 October 2025

**Sent for Review** 29 October 2025

**Reviewed preprint posted** 05 January 2026

**Reviewed preprint revised** 05 March 2026

**Version of Record published** 07 April 2026

**Abstract** Antibodies are encoded by nucleotide sequences that are generated by V(D)J recombination and evolve according to mutation and selection processes. Existing antibody language models, however, focus exclusively on antibodies as strings of amino acids and are fitted using standard language modeling objectives such as masked or autoregressive prediction. In this paper, we first show that fitting models using this objective implicitly incorporates nucleotide-level mutation processes as part of the protein language model, which degrades performance when predicting effects of mutations on functional properties of antibodies. To address this limitation, we devise a new framework: a deep amino acid selection model (DASM) that learns the selection effects of amino acid mutations while explicitly factoring out the nucleotide-level mutation process. By fitting selection as a separate term from the mutation process, the DASM exclusively quantifies functional effects: effects that change some aspect of the function of the antibody. This factorization leads to substantially improved performance on standard functional benchmarks. Moreover, our model is an order of magnitude smaller and multiple orders of magnitude faster to evaluate than existing approaches, as well as being readily interpretable.

## Introduction

Antibodies are remarkable molecules that can bind essentially any target with high affinity and specificity. Natural antibodies are generated through V(D)J recombination and refined by somatic hypermutation (SHM) and affinity-based selection in germinal centers. Antibodies are also important drugs, and improving binding and other important properties is thus an area of active research. A major goal

in the field is to predict the effect of changing one amino acid for another at a given site of an antibody, both for antibody engineering and for understanding naturally occurring antibodies.

Antibody 'foundation' language models are trained on large datasets of naturally occurring antibody sequences to assist with this and related problems (*Ruffolo et al., 2021*; *Leem et al., 2022*; *Barton et al., 2024a*; *Kenlay et al., 2024*; *Turnbull et al., 2024*; *Singh et al., 2025*; *Huh et al., 2025*). These models are trained using the masked objective, in which a site is masked from an antibody amino acid sequence, and the model is trained to predict the masked amino acid given the remaining sequence. Recent models of this type have hundreds of millions or billions of parameters and are trained on around a billion sequences (*Barton et al., 2024b*).

While the masked objective has been very useful for modeling human language (*Devlin et al., 2018*), this approach may not be ideal for learning functional effects of mutations to antibody sequences. To understand why, it is useful to consider: what could a language model learn in order to succeed under the masked modeling objective? It could memorize the germline genes (*Olsen et al., 2024*; *Ng and Briney, 2025*) and learn about the probabilities of V(D)J recombination. It could learn the codon table, as according to this table some amino acid mutations are much more likely than others. It could learn rates of SHM, because codons containing mutable nucleotide sites are more likely to deviate from germline than those in sites that are less mutable (*Sheng et al., 2017*). It could also learn about the impact of amino acid mutations on antibody function through natural selection in the course of affinity maturation, which is the desired signal (*Chungyoun et al., 2024*). However, this desired signal is confounded by the preceding factors.

In this paper, we first demonstrate that masked language models learn all these factors shaping antibody sequences, despite most being irrelevant for functional prediction. Indeed, we show that conflating mutation and selection processes degrades performance on functional prediction tasks. We then develop a model that explicitly accounts for phylogenetic relationships, the codon table, and SHM patterns, allowing it to focus exclusively on functional effects.

Our approach separates mutation and selection processes by encoding functional effects in a deep amino acid selection model (DASM) while explicitly modeling mutation using a separate fixed model trained on neutrally evolving data (*Sung et al., 2025a*). This fixed model uses convolutions on 3-mer embeddings to deliver wide context sensitivity without needing a large number of parameters: the variant we use has around the same number of parameters as the classic S5F (*Yaari et al., 2013*) 5-mer model. The DASM, trained on substantially less data, outperforms AbLang2 (*Olsen et al., 2024*) and general protein language models including ESM2 (*Rao et al., 2020*; *Rives et al., 2021*) and ProGen2-small (*Nijkamp et al., 2023*). This outperformance holds on the largest benchmark datasets of the FLAb collection (*Chungyoun et al., 2024*) and on recent high-throughput binding assays (*Petersen et al., 2024*; *Kirby et al., 2025*). Unlike existing models, DASMs process complete sequences in a single pass and directly output selection factors for all possible mutations. DASMs are thus orders of magnitude faster than existing models, enabling intensive use on a laptop without a GPU. We provide a paired (heavy and light chain) model with open weights and associated code as part of our `netam` package https://github.com/matsengrp/netam; *Matsen et al., 2025a*.

## Results
### Antibody language models are biased by nucleotide-level mutation processes

We first sought to understand the extent to which processes such as neutral mutation rate and the codon table influence antibody language model prediction at masked sites. To do so, we used AbLang2 (*Olsen et al., 2024*) as a case study. It has been implemented with more consideration of biology than other models, as it distinguishes between germline-encoded and non-germline-encoded sites in its training. To start, we examined a single naive BCR sequence obtained from a recent study that performed deep sequencing of human antibody repertoires (*Rodriguez et al., 2023*). We iterated over each site in this antibody's protein sequence, masked the site's amino acid, and used AbLang2 to compute the likelihood of each of the 19 alternative amino acids at that site, then normalized to get a probability. Next, using the antibody's nucleotide sequence, we split the alternative amino acids at a given site into two groups: those that can be encoded by a codon a single nucleotide change away from the original codon, and those that require multiple mutations.

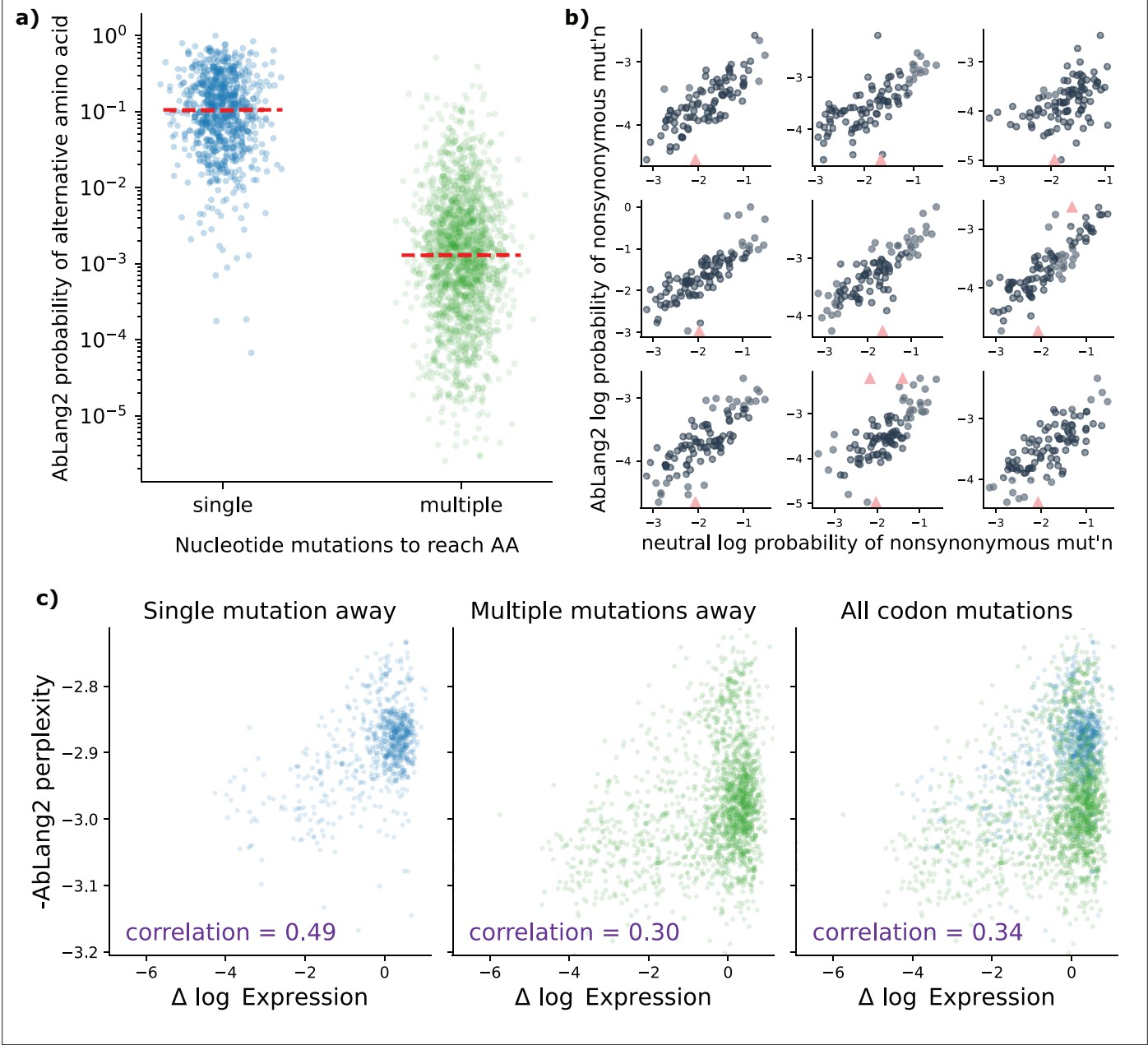

**Figure 1.** Nucleotide-level mutation processes distort protein language model predictions. (**a**) AbLang2 assigns almost 100× lower probabilities to amino acids requiring multiple nucleotide mutations compared to single-mutation variants. Each point represents a possible amino acid substitution at a single site in the amino acid sequence. (**b**) AbLang2 probabilities correlate with neutral somatic hypermutation probabilities across the V-encoded portion of nine naive sequences, demonstrating how the model is strongly impacted by mutation bias. Each point represents a site in the sequence. Triangles are outliers that have been brought into the y range. (**c**) AbLang2 functional prediction accuracy drops substantially for amino acids that are multiple (2 or 3) nucleotide mutations away from the wild-type codon. Data are from *Koenig et al., 2017*.

The online version of this article includes the following figure supplement(s) for figure 1:

**Figure supplement 1.** Conflating mutation and selection hinders functional prediction.

We found a striking difference in the AbLang2 amino acid probabilities (*Figure 1a*). The median probability for amino acids requiring multiple mutations to a codon was almost two orders of magnitude lower compared to those that only required one. This bias is consistent with the hypothesis that the AbLang2 predictions are influenced by mutation processes unrelated to selection.

The bias can be explained by the masked training procedure. It has already been established that language models trained with the masked objective memorize germline sequences (*Olsen et al., 2024*; *Ng and Briney, 2025*). During SHM, single-nucleotide-per-codon mutations to the germline sequence are much more likely than multi-nucleotide codon mutations. As a result, when predicting relative probabilities of alternative amino acids, language models would be expected to assign much higher probabilities to alternative amino acids that only require a single-nucleotide codon mutation (*Figure 1a*).

Given this observation, we also hypothesized that AbLang2 probabilities are influenced by differences in the rate of SHM between sites. SHM is a purpose-built enzymatic process that introduces mutations into BCR-coding DNA (*Pilzecker and Jacobs, 2019*), which occurs independently of the process of natural selection in the germinal center. The rate of SHM varies by an order of magnitude or more between sites, and these biases are well characterized using probabilistic models (*Yaari et al., 2013*; *Sung et al., 2025a*). To test our hypothesis, we examined nine arbitrarily selected sequences from the *Rodriguez et al., 2023*, dataset. For each sequence, we used a recent model of SHM (*Sung et al., 2025a*) to compute per-nucleotide-site rates, and then used the strategy from our previous paper (*Matsen et al., 2025b*) to convert these rates into per-codon-site probabilities of nonsynonymous mutations.

We found a clear correlation between these neutral SHM probabilities and probabilities of mutations estimated using AbLang2 (*Figure 1b*). The Wald test for a nonzero slope reported a p-value below machine precision in each case.

We next sought to understand if these biases distorted predictions on a functional prediction task. To do so, we used the largest dataset (*Koenig et al., 2017*) of the FLAb (*Chungyoun et al., 2024*) collection of benchmarks, which measures the effect of single mutations on expression in a phage display library. We scored each single-mutation variant using a score that, up to a linear transformation, is the difference between the log probability the model assigns to the variant amino acid and the log probability it assigns to the wild-type amino acid at that position (see masked-marginals pseudo-perplexity in Materials and methods). We then computed the Pearson correlation between these AbLang2 scores and the experimentally measured expression effects, splitting by number of nucleotide mutations per codon as before.

We found a significant drop (from a correlation of 0.49 to 0.30) in predictive performance when going from amino acids that required one mutation to those that required multiple mutations (*Figure 1c*). Furthermore, when we considered all amino acids together, the correlation remained at 0.34. When we colored these data points by their probability of mutation under an SHM process (*Figure 1—figure supplement 1*), we can clearly see that the data points are spread out by the AbLang2 model according to their SHM rate, hindering functional prediction.

In summary, we found that nucleotide-level effects hamper the ability of AbLang2 to predict functional effects of mutations. We then developed an alternative modeling framework that directly factors out nucleotide-level effects.

## Fitting a DASM

We implemented a model to learn amino acid preferences of antibodies without being influenced by germline genes, the codon table, or SHM biases (*Figure 2*). To do so, we extended our previous work estimating a single selection value for every site (*Matsen et al., 2025b*) to estimating a value for every alternative amino acid at every site. This extension is analogous to going from a dN/dS type estimate (*Goldman and Yang, 1994*; *Muse and Gaut, 1994*) to a sitewise mutation-selection model (*Halpern and Bruno, 1998*). Our previous model only operated on antibody heavy chains. This new model operates on heavy chains, light chains, or heavy-light pairs.

As in this previous work, we performed clonal-family clustering, phylogenetic inference, and ancestral sequence reconstruction to generate collections of nucleotide 'parent-child pairs' or PCPs (*Figure 2a*). This resulted in around 2 million PCPs that were used for training (*Table 1*). Instead of predicting the likelihood of nonsynonymous mutations, we predicted the likelihood of codon mutations using a deep-learning analog of a mutation-selection model. The neutral-probability model was inferred separately using out-of-frame data (*Sung et al., 2025a*). We also added a 'multihit correction' (see Materials and methods) to this neutral model which accounts for the spatial clustering of SHM (*Spisak et al., 2020*), resulting in an elevated probability of multiple mutations in a given codon.

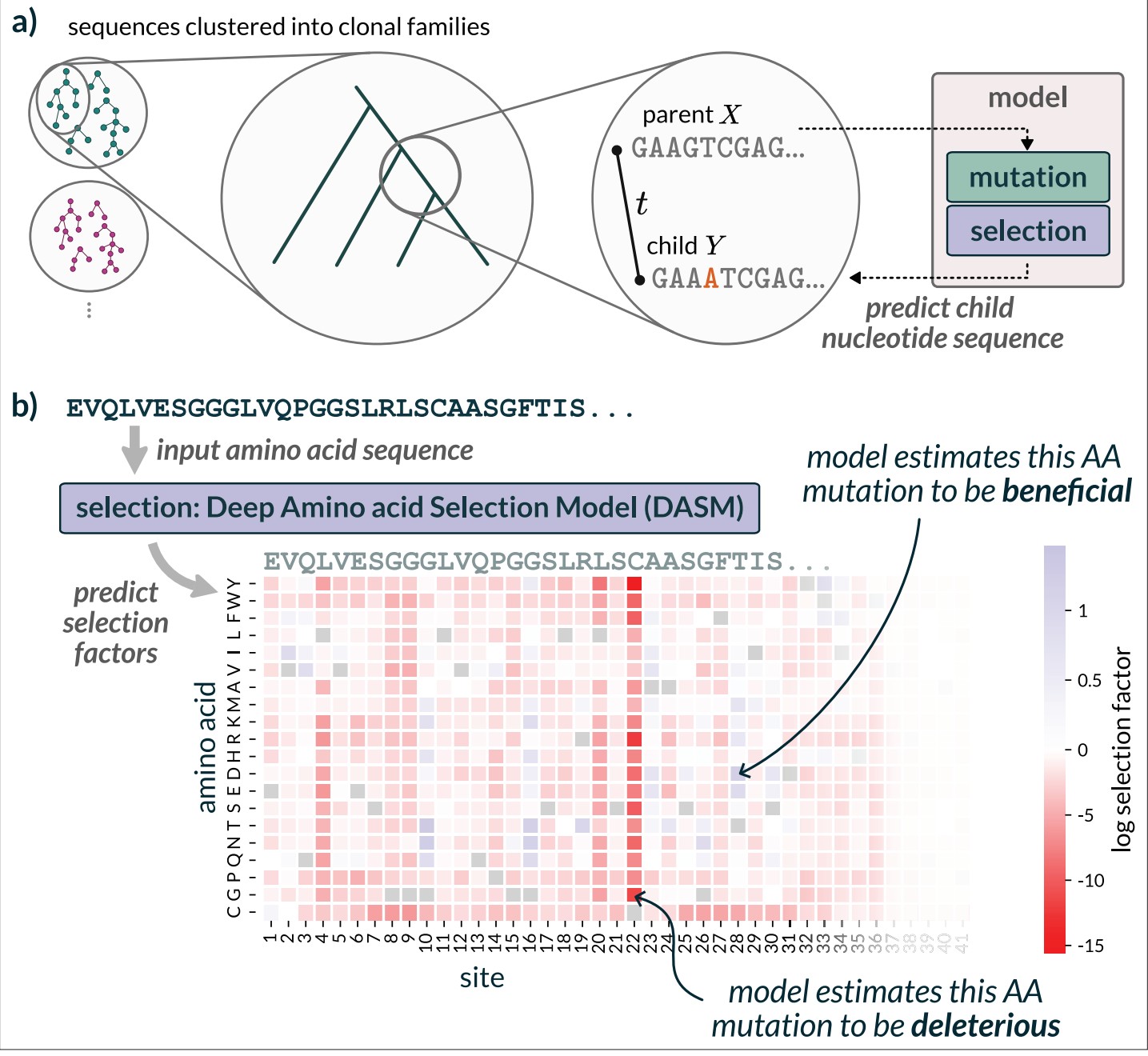

**Figure 2.** Our model separates mutation from selection to predict functional effects without nucleotide-level biases. (**a**) Our model combines a fixed mutation component (trained on non-functional data) with a learned selection component (deep amino acid selection model [DASM] transformer). Training uses inferred parent-child sequence pairs from reconstructed B cell phylogenies to predict natural affinity maturation after a jointly inferred time $t$. (**b**) The DASM directly predicts selection factors for all amino acid substitutions at every position in a single forward pass. Positive factors indicate beneficial changes, and negative factors indicate deleterious changes.

The online version of this article includes the following figure supplement(s) for figure 2:

**Figure supplement 1.** Overview of full detailed methods.

**Figure supplement 2.** Multihit correction improves model fit.

The selection component of the model estimates per-site per-amino-acid selection factors given an input amino acid sequence via a transformer-encoder (*Figure 2b*). We trained DASMs of several sizes (~1M, ~4M, ~7M) using joint optimization of branch length $t$ and parameters of the DASM (see Materials and methods for details). We found that the ~4M parameter model performed the best

**Table 1.** Correlation of models with predicting effects of single mutations on antibody expression and antigen binding, as measured in *Koenig et al., 2017*.

Masked language models scored using per-sequence pseudo-perplexity following the FLAb protocol (*Chungyoun et al., 2024*).

| Model | Binding | | Expression | |
|---|---|---|---|---|
| | Heavy | Light | Heavy | Light |
| AbLang2 | −0.114 | −0.108 | 0.153 | −0.109 |
| DASM | **0.335** | **0.316** | **0.688** | **0.674** |
| ESM2 | 0.009 | 0.243 | 0.384 | 0.416 |
| ProGen2 | 0.156 | 0.276 | 0.559 | 0.568 |

according to our objective function. This DASM has 8 attention heads, 32 dimensions per head, a feedforward dimension of 1024, 5 transformer layers, and a dropout probability of 0.1; we will use this version for the rest of the paper. Hyperparameters other than model size were carried over from our previous work (*Matsen et al., 2025b*).

As an initial comparison between DASM and AbLang2, we used DASM to predict selection factors for the unmutated antibody sequence used in the Koenig benchmark *Koenig et al., 2017*; we found that the selection factors were predictive of functional measurements irrespective of the number of mutations per codon (*Figure 3a*, *Figure 3—figure supplement 1*). We were surprised by the strength of the correlation for amino acid mutations that require multi-nucleotide codon mutations, given that training signal should be weaker for those rarer mutations. We also found that the selection factors showed very similar distributions for amino acid mutations requiring single- vs. multi-nucleotide codon mutations (*Figure 3—figure supplement 2*). Furthermore, the DASM selection factors captured the alternating pattern of selective constraint on beta sheets evident in the expression data (red columns in *Figure 3b*, *Figure 3—figure supplement 3*).

## Thrifty+DASM accurately predicts affinity maturation

Next, we evaluated the ability of the DASM to predict the course of natural affinity maturation. To do so, we used PCPs from the Rodriguez data (*Table 1*), which were not used for training the DASM, nor for training AbLang2. For each PCP, we quantified each model's ability to predict both the location of nonsynonymous mutations in the child sequence and the identity of the mutant amino acid. For the DASM, we computed a mutation's probability by multiplying the mutation's selection factor, predicted by the DASM, with its neutral mutation probability, predicted using the same fixed neutral mutation model used to train the DASM (*Sung et al., 2025a*).

We found that the DASM-based approach was better than AbLang2 at predicting the location of nonsynonymous mutations observed in the PCPs (*Figure 3—figure supplement 4a and b*). In fact, the overlap between observed and expected is substantially better than what we were able to obtain in a highly controlled mouse experiment (*DeWitt et al., 2025*) with a customized mutation model and a deep mutational scan that quantified how mutations affect binding to the relevant antigen (*Johnson et al., 2025*).

We also found that the DASM-based approach was better than AbLang2 at predicting the identities of mutant amino acids. To compare the models at this task, we used a notion of 'conditional perplexity'. Specifically, for each PCP where the parent and child sequences differ on the amino acid level, we calculated the perplexity of the residues in the child that are different from the parent, conditioning probabilities on there being a substitution (see cond-ppl definition in Materials and methods). We do this to give AbLang2 the best chance of succeeding: although it might have difficulty predicting the occurrence of mutations (*Figure 3—figure supplement 4a, b*), it might still be able to predict the identity of mutations.

Indeed, the DASM had lower conditional perplexity than AbLang2 on 1000 sequences from the *Rodriguez et al., 2023*, dataset (*Figure 3—figure supplement 4c*). The median value for DASM was 4.88, compared to 7.39 for AbLang2. In addition to having a lower median, the DASM also has fewer very large values.

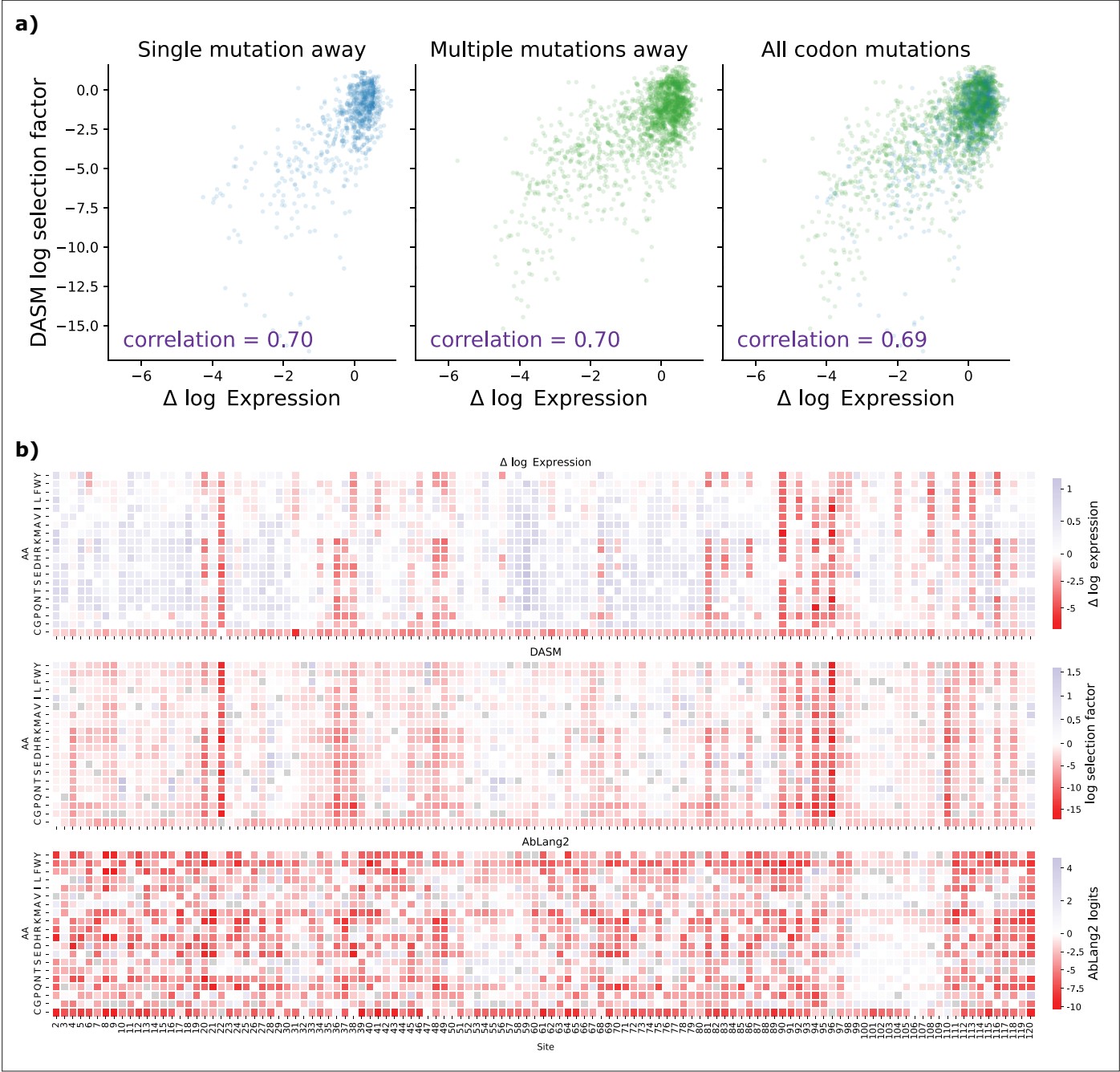

**Figure 3.** Comparing model predictions with experimentally measured effects of mutations on antibody expression from *Koenig et al., 2017*. (**a**) The deep amino acid selection model (DASM) maintains high predictive accuracy on functional effects of mutations regardless of codon accessibility. The correlation is equally high for amino acid mutations that only require a single-nucleotide mutation (left plot) vs. amino acid mutations that require multi-nucleotide mutations (center plot), demonstrating successful separation of mutation bias from functional effects. Compare *Figure 1c*. (**b**) DASM predictions mimic patterns in the expression data. For additional heatmap comparisons, see *Figure 3—figure supplement 3*.

The online version of this article includes the following figure supplement(s) for figure 3:

**Figure supplement 1.** Light chain version of main figure.

**Figure supplement 2.** Comparison of selection factors for codon neighbors in the deep amino acid selection model (DASM).

**Figure supplement 3.** Heavy chain heatmap.

**Figure supplement 4.** Parent-child pair (PCP) predictions.

**Figure supplement 5.** Scatterplots for Shanehsazzadeh data.

**Figure supplement 6.** MAGMA scatter plots.

## The DASM outperforms masked and autoregressive models at predicting functional effects of mutation

We next performed a more comprehensive characterization of the ability of models to predict mutational effects. Because the DASM is trained exclusively on evolutionary trajectories rather than functional measurements, evaluation on expression and binding benchmarks is strictly zero-shot with no risk of data leakage. We expanded our suite of comparator models to include ESM2 (*Rao et al., 2020*; *Rives et al., 2021*), a general protein language model trained with a masked objective, and ProGen2 (*Nijkamp et al., 2023*), an autoregressive language model. We used the small general variant of ProGen2, which the FLAb benchmarking paper found to be the best overall model (*Chungyoun et al., 2024*), and the 650M variant of ESM2. To predict the plausibility of mutant sequences under these models, we followed the FLAb protocol and calculated each sequence's pseudo-perplexity for AbLang2 (*Olsen et al., 2024*) and ESM2 (*Lin et al., 2023*), scoring each variant in its own sequence context (Materials and methods), and each sequence's perplexity for ProGen2, then compared the model outputs to experimentally measured mutational effects using Pearson correlation. To predict mutational effects using the DASM, we simply fed the DASM with the unmutated sequence from the deep mutational scanning (DMS) experiment and used the output selection factors as predicted effects. We cannot use perplexity because selection factors alone do not give likelihoods.

We based our evaluations on the April 2024 version of the FLAb benchmarking project (*Chungyoun et al., 2024*), which preceded our work and thus was not subject to selection bias by us. We also benchmarked high-throughput binding data (more recent than FLAb) from the Whitehead lab (*Petersen et al., 2024*; *Kirby et al., 2025*) that provided affinity measurements across many variants and antigens.

To start, we evaluated these models on the two largest datasets from the FLAb collection. The first dataset, from *Koenig et al., 2017*, is a DMS experiment performed on the Fab of the anti-VEGF antibody G6.31. The authors generated single-site saturated mutagenesis libraries incorporating all possible amino acid mutations to both the variable heavy and variable light domains. Using phage display, they then performed two independent selection steps: one in which they selected for Fab variants that were stably expressed, and another in which they selected for Fab variants that could bind VEGF. They quantified mutational effects on these properties by computing enrichment ratios of each mutation in selected vs. unselected pools, as quantified using deep sequencing.

We found that the DASM outperformed the other models on both binding and expression (*Table 1*, *Figure 3—figure supplement 3*). Scaling ESM2 from 650M to 3B parameters did not improve performance: indeed, the larger model showed slightly degraded correlations, particularly for light chain predictions (*Table 2*). This is consistent with recent observations that medium-sized protein language models can outperform larger ones on transfer learning tasks (*Vieira et al., 2025*). Thus, we only used the 650M model in the rest of our analyses.

We next turned to the second-biggest dataset of the FLAb benchmark, from *Shanehsazzadeh et al., 2023*. In this study, the authors took an anti-HER2 antibody, trastuzumab, and used deep generative models to redesign the heavy chain complementarity determining regions (HCDRs). They generated libraries of HCDR variants, including both HCDR3-only designs and HCDR123 designs. Unlike the first dataset described above, many variants had multiple mutations relative to the unmutated antibody sequence. The experiments expressed antibody variants in *Escherichia coli* and sorted by FACS based on binding signal. We restricted our attention to sequences with the two most common heavy chain amino acid lengths (119 and 120) as the other lengths did not have enough data to make a full comparison.

We assessed the ability of models to predict these binding measurements. For the language models, we computed the perplexity of each variant, as before. For the DASM, we calculated a consensus

**Table 2.** Correlation of models with binding measurements on the data of *Shanehsazzadeh et al., 2023*, which typically involves multi-mutant variants.
See *Figure 3—figure supplement 5* for scatterplots.

| Seq. length | AbLang2 | DASM | ESM2 | ProGen2 |
|---|---|---|---|---|
| 119 | 0.263 | **0.458** | 0.191 | 0.074 |
| 120 | 0.166 | **0.518** | 0.308 | 0.052 |

**Table 3.** Correlations between model predictions and binding affinity.

The 'Petersen' data is from an experiment probing the rules of recognition for influenza neutralizing antibodies (*Petersen et al., 2024*) and 'Kirby' data is from combinatorial libraries applying combinations of mutations on the path from naive to mature SARS-CoV-2 antibodies (*Kirby et al., 2025*). See *Figure 3—figure supplement 6* for corresponding scatter plots.

| Source | Antibody | AbLang2 | DASM | ESM2 | ProGen2 |
|---|---|---|---|---|---|
| | 222-1C06 | 0.041 | **0.248** | −0.039 | −0.030 |
| Petersen (*Petersen et al., 2024*) | 319-345 | 0.079 | **0.279** | 0.199 | 0.222 |
| | 002-S21F2 | −0.246 | **−0.165** | −0.283 | −0.353 |
| | Ab_2-15 | −0.325 | **0.094** | −0.265 | −0.069 |
| Kirby (*Kirby et al., 2025*) | C118 | −0.624 | **0.276** | −0.293 | −0.159 |

sequence for each sequence length, and then used the DASM to compute selection factors for that sequence. Then, we computed a score for each variant by summing the log selection factors for each amino acid mutation in the variant relative to the consensus sequence. As above, the resulting correlation is substantially higher for the DASM compared to the other models (*Table 2*).

For our third validation, we sought to see how our model could predict affinity in the context of large-scale yeast display experiments using MAGMA-seq (*Petersen et al., 2024*). We were especially interested in this assay because the measurement is affinity-only rather than a conflation of affinity and expression as in the previous datasets. Because none of these models are trained in an epitope-specific way, this will be a challenge for the models. Specifically, MAGMA-seq determines quantitative monovalent binding affinities through deep sequencing of barcoded Fab libraries sorted at multiple antigen concentrations in a manner analogous to Tite-Seq (*Adams et al., 2016*). These data are not present in the FLAb benchmarks. We used the largest datasets in the original MAGMA-seq paper (*Petersen et al., 2024*), which test variants of influenza antibodies. We also used MAGMA-seq data from *Kirby et al., 2025*, which combinatorially applied mutations from development trajectories of mature human antibodies against SARS-CoV-2 spike protein.

We found that the DASM performed substantially better than other models for this task (*Table 3*, *Figure 3—figure supplement 6*). Nearly all correlation coefficients were small, pointing to the difficulty in predicting effects of mutations on antigen-specific binding affinities. Nevertheless, the DASM was the only model to have a positive correlation for most datasets. The UCA002-S21F2 lineage proved challenging for all models, with all models having a negative correlation. This may not be surprising given that 002-S21F2 uses a rare VH5-51/VK1-33 gene combination found in only 3 out of 5252 SARS-CoV-2 antibodies in the CoV-AbDab database (*Kumar et al., 2022*).

## DASMs are orders of magnitude faster than competing models and much smaller

We next compared the computational requirements for doing these evaluations. We imagined two settings: one where a scientist wishes to evaluate a small collection of sequences on their laptop without a GPU, and the other where they wish to evaluate a larger collection of sequences on a GPU server. Our timing script ran the DASM, as well as AbLang2 and ESM2, using stepwise masking to obtain amino acid likelihoods. The local machine was a MacBook Pro with an M2 Max chip and 96 GB of memory. The GPU server had an AMD EPYC 75F3 32-core processor with 995 GB of memory and an NVIDIA A100 80 GB GPU.

**Table 4.** Computational efficiency comparison on sequences from the MAGMA-seq experiments.

10 sequences were run on CPU, and 100 on the GPU server.

| Model | CPU (s/seq) | GPU (s/seq) |
|---|---|---|
| DASM | 0.0097 | 0.0053 |
| AbLang2 | 11.0529 | 0.5196 |
| ESM2 | 112.5989 | 7.6090 |

The DASM achieved dramatic computational efficiency gains compared to masked language models (*Table 4*). On a CPU, the DASM evaluates sequences over 1000 times faster than AbLang2 and more than 10,000 times faster than ESM2. On a GPU, the DASM is 100 times faster than AbLang2 and over 1000 times faster than ESM2. This speed advantage comes from the DASM providing predictions for all variants in a single pass, eliminating the need for iterative masking procedures required by masked language models. (We note that it is possible to make predictions from masked language models without masking, although this is not considered best practice because of the influence of the unmasked residue.)

This evaluation was actually generous to the masked language models, as it assumes that we are evaluating likelihoods of masked sites and combining them to evaluate multi-mutant variants. That is in contrast to what was actually done in the performance evaluation, which is direct evaluation of the sequence perplexity for each sequence individually (which should be more accurate). ProGen2 was not included in this comparison because it is categorically slower for this use case: autoregressive models for a panel of variants require running each variant individually through the model.

The DASM benchmarked here is also significantly smaller than alternate models. It has 4M parameters, compared to 45M for AbLang2, 650M for ESM2, and 151M for ProGen2. We suspect that this efficiency is enabled by DASM learning selection effects only, rather than selection and mutation-level effects as for existing models.

In summary, we imagine end-users will find it convenient to download a small (23 MB) weights file and run the DASM on their laptop, as compared to existing models requiring more powerful hardware.

## Discussion

We have motivated and developed a new direction for protein language modeling. To motivate this approach, we showed how masked language modeling implicitly learned the codon table and SHM rates, properties that are orthogonal to antibody function. This incorporation degrades model performance for predicting effects of mutations on antibody expression and binding.

To address this limitation, we introduced a new framework: a DASM, which models selection effects separately from mutation effects. By modeling these components independently, the DASM more accurately predicts the functional consequences of amino acid mutations while having dramatically reduced computational requirements. The success of our approach highlights the importance of incorporating biological realities into machine-learning models for protein engineering. This aligns with recent calls for deeper dialogue between machine learning and evolutionary biology to address phylogenetic biases in biological foundation models (*York and Avasthi, 2025*).

Thrifty+DASM can be viewed as a neural network extension of the models rooted in the work of *Halpern and Bruno, 1998*. There, the probability of a codon mutation is expressed as the product of that mutation under a neutral nucleotide process, times a term representing the selection for or against the corresponding amino acid substitution at that site. Although the original model expressed selection coefficients in a parameter-sparse way using equilibrium frequencies, further model elaborations allowed for per-site-per-amino acid selection factors, as described here, either in a fixed-effects (*Tamuri et al., 2012*; *Tamuri et al., 2014*) or random-effects (*Rodrigue and Lartillot, 2014*) framework. Other related work includes developing evolutionary models using DMS experiments (*Bloom, 2017*; *Hilton et al., 2017*) and models of sequence-structure compatibility (*Robinson et al., 2003*). Thrifty+DASM also extends our previous work that estimates a single value of natural selection at every site (*Matsen et al., 2025b*); such a model is not comparable to an antibody foundation model because it does not estimate per-amino-acid probabilities.

By separating out selection from mutation and the effect of evolutionary contingency, DASMs provide direct interpretability: the selection factors indicate which amino acid mutations are beneficial or deleterious at each position. This interpretability could prove valuable for antibody engineering, where understanding which specific mutations would drive improved expression or binding is crucial for rational design. We have dedicated a companion paper to leveraging this interpretability to provide new perspectives on the operating rules of affinity maturation (*Matsen et al., 2025b*): that work provides a nuanced sitewise perspective on natural selection in antibodies that challenges classical oversimplified views of selection patterns. For users wishing to explore the relationship between DASM selection factors and structure for human antibodies in the SAbDab database (*Dunbar et al.,*

*2014*), we have made interactive 3D visualizations of DASM selection factors using `dms-viz` (*Hannon and Bloom, 2024*) available at https://matsen.group/dasm-viz/v1/.

The current generation of DASM model does not use any antigen-labeled training data. The signal that it leverages to infer some limited ability to predict binding comes from natural affinity maturation. This affinity maturation comes through natural repertoires and so represents a mix of all of the antigens to which the sampled individuals have been exposed.

In future work, we will explore fine-tuning of these models with antigen-specific data. While DASM provides better-than-state-of-the-art zero-shot performance on binding benchmarks, recent work for binding focuses on using protein embeddings as inputs to subsequent classifiers (*Wang et al., 2024*) or fine-tuning for binding prediction (*Wang et al., 2025*; *Barton et al., 2024b*). Because the DASM framework separates mutation from functional properties, a fine-tuned DASM may improve on these previous efforts. We also have plans to extend the DASM framework to estimate the effect of natural selection on insertion and deletion events.

We are beginning the process of extending DASMs to other settings. Although antibodies form an interesting first application of DASMs and have many properties that make DASMs a useful tool, DASMs are not restricted to the antibody case. Viral sequences also exist in abundance and may be amenable to the DASM approach. Rather than many clonal families for antibodies, we will fit a DASM with many separate alignments for related but distinct evolutionary histories.

Taking this idea to its logical extent, we would like to train a DASM model analogous to ESM (*Lin et al., 2023*) for all proteins. However, this will be a major computational undertaking. Even setting training such a model aside, doing phylogeny and ancestral sequence reconstruction on all protein alignments will require careful planning. If we followed the steps used here for phylogeny and ancestral sequence reconstruction on the 2.6 million sequence alignments used for the MSA Transformer (*Rao et al., 2021*), it would take around 3000 CPU-years.

DASMs represent a new paradigm for deep models of protein function that leverage evolutionary information directly. Even in this initial implementation, DASMs outperform existing foundation models while requiring orders of magnitude less computation, parameters, and training data. This efficiency suggests that evolutionary structure provides a powerful approach for understanding protein function.

## Materials and methods
### Data
BCR sequence data was processed with partis (*Ralph and Matsen, 2022*) to cluster into clonal families and infer germlines. Inferred insertions or deletions were reversed, so that all sequences align to the naive sequence without gaps. We selected clonal families with at least two productive sequences; a sequence is considered productive if the canonical cysteine and tryptophan codons that flank the CDR3 are in the same frame as the start of the V segment (although they can be mutated), and there are no stop codons. We excluded sequences with stop codons. Following the training of other LLMs (e.g. *Olsen et al., 2024*), we also excluded sequences with mutated conserved 'signature' cysteines, in contrast to our previous work (*Matsen et al., 2025b*).

As in our previous work, tree inference and ancestral sequence reconstruction were performed per clonal family with the K80 substitution model using the naive sequence as outgroup, allowing mutation rate heterogeneity across sites using a four-category FreeRate model using IQ-Tree (*Minh et al., 2020*). However, for paired data, we used the *edge-linked-proportional* partition model in IQ-Tree that allowed the heavy and light chains to evolve at overall different rates (*Chernomor et al., 2016*). Because these clonal families are independent, these phylogenetic inferences were run in parallel.

Once this was done, we had a set of PCPs that correspond to the pairs of parent and child sequences on the edges of the phylogenetic tree (*Figure 2a*). We used these PCPs to train the model.

We denote pairs of parent and child sequences as $(X, Y)$, where $X$ is the parent sequence and $Y$ is the child sequence. We use $\bar{X}$ to denote the amino acid sequence corresponding to $X$.

## Model

### Formulation and loss

Assume we are given a parent codon sequence $X$ and a child sequence $Y$. We will use $\bar{c}$ to denote the amino acid sequence of codon $c$, and use $\bar{Z}$ to denote the amino acid sequence of codon sequence $Z$. We will use $j$ to denote codon sites. As before (**Matsen et al., 2025b**), we model the neutral probability of a mutation to codon $c$ as the product of per-nucleotide-site mutation probabilities: specifically, the probability of mutating from the parent codon to codon $c$ is the product of the probabilities of each constituent nucleotide mutation (or non-mutation) under the neutral model. This results in the probability $p_{j,c}(t, X)$ of a mutation to codon $c$ at site $j$ after time $t$.

The selection term $f_{j,\bar{c}}(\bar{X})$ is a 'selection factor' that quantifies the natural selection happening at site $j$ if it was to be changed to $\bar{c}$. If this value is greater than 1, it predicts that the mutation would be beneficial in the course of affinity maturation, while if it is less than 1, it predicts that the mutation would be deleterious.

We parameterize the DASM $f$ using the standard transformer-encoder architecture (**Vaswani et al., 2017**): an amino acid embedding, sinusoidal positional encodings, and PyTorch's `TransformerEncoder` module. The only non-standard component to this architecture is a custom 'wiggle' activation function to the output layer that prevents extreme selection factors as previously described (**Matsen et al., 2025b**). This function asymptotes to zero for highly deleterious mutations and grows sublinearly for beneficial ones.

We will now define the likelihood $\ell_{j,c}(t, X)$ of codon $c$ at site $j$ given time $t$ and parent sequence $X$. When $c$ is a non-wild-type codon, it is

$$\ell_{j,c}(t, X) := \begin{cases} p_{j,c}(t, X) f_{j,\bar{c}}(\bar{X}) & \text{if } c \text{ codes for an amino acid} \\ 0 & \text{if } c \text{ codes for stop} \end{cases} \tag{1}$$

where $f_{j,a}(\bar{X}) := 1$ when $a$ is the amino acid for the wild-type codon. We then take the likelihood of the wild-type codon to be 1 minus the sum of these non-wild-type codons.

The overall likelihood for a PCP is then

$$\prod_j \ell_{j,y_j}(t, X)$$

where $y_j$ is the $j$th codon of $Y$. We optimize this likelihood jointly over the branch lengths $t$ and parameters of the selection model $f$. This joint optimization is performed cyclically, in which a complete cycle consists of neural network optimization followed by branch length optimization for every PCP. The parent sequence and the child sequence are pre-estimated, fixed, and used as training data. The branch lengths are independent and so are optimized in parallel.

Because $\ell_{j,c}(t, X) = p_{j,c}(t, X)$ when $c$ codes for the wild-type amino acid, this approach effectively fixes the selection factors for neutral substitutions to be 1 for the purposes of branch length optimization. This is useful because it gives us a 'gauge' that eliminates a potential unidentifiability in the model.

Especially early in training, it can happen that $p_{j,c}(t, X) f_{j,\bar{c}}(\bar{X})$ is greater than one, or that the sum of such terms is greater than one. In these cases, the sum is clamped to be a little less than 1.

### Perplexity

Perplexity is the standard way of evaluating the plausibility of a sequence according to a model: it is the across-site geometric mean of the inverse probability of the observed amino acid. These inverse probabilities can be interpreted as the effective number of tokens that the model considers plausible at each position. This is equivalent to the exponential of the average negative log probability of each token given its preceding context:

$$\text{ppl}(x) = \exp\left(-\frac{1}{n} \sum_{i=1}^{n} \log p\left(x_i \mid x_{<i}\right)\right),$$

where $x = (x_1, x_2, \ldots, x_n)$ is a sequence of $n$ tokens, and $x_{<i}$ represents all tokens preceding position $i$.

For encoder-only models like ESM2 and AbLang2, which mask tokens rather than using autoregressive prediction, we follow the authors of these models and instead calculate a 'pseudo-perplexity' using the probability of each token given all other tokens in the sequence:

$$\text{pseudo-ppl}(x) = \exp\left(-\frac{1}{n}\sum_{i=1}^{n}\log p\left(x_i \mid x_{\setminus i}\right)\right),$$

where $x_{\setminus i}$ denotes the sequence with token $i$ masked out.

An alternative 'masked-marginals' approach scores variants using only wild-type context. For a wild-type sequence $w$, masked-marginals computes $p(a \mid w_{\setminus i})$ for all amino acids $a$ at each position $i$ once, then uses these wild-type-derived probabilities to compute pseudo-perplexity for any variant $x$:

$$\text{mm-pseudo-ppl}(x) = \exp\left(-\frac{1}{n}\sum_{i=1}^{n}\log\left(p\left(x_i \mid w_{\setminus i}\right)\right)\right).$$

For a single-mutation variant $x$ that differs from wild-type $w$ only at position $j$, all terms except position $j$ cancel when comparing to wild-type, giving

$$\log(p(x_j \mid w_{\setminus j})) - \log(p(w_j \mid w_{\setminus j}))$$

$$= n\left[\log(\text{mm-pseudo-ppl}(w)) - \log(\text{mm-pseudo-ppl}(x))\right].$$

Thus, the log-probability difference between variant and wild-type amino acids equals, up to an additive constant $n\log(\text{mm-pseudo-ppl}(w))$ that depends only on the wild-type sequence, negative $n$ times the log pseudo-perplexity of the variant.

For *Figure 1c* on the single-mutant Koenig dataset, we found that this approach gave a substantially higher correlation for AbLang2 and so used it in that figure. For benchmarking comparisons (*Table 1*), we followed standard practice and used per-sequence pseudo-perplexity.

We also use a notion of 'conditional' perplexity: for each PCP $(x, y)$ of sequences that differ on the amino acid level, calculate the perplexity of the residues that are different from the parent, conditioning probabilities on there being an amino acid substitution. That is, if we let $S_{x,y}$ be the sites that differ between $x$ and $y$, then

$$\text{cond-ppl}(x, y) = \exp\left(-\frac{1}{|S_{x,y}|}\sum_{i \in S_{x,y}}\log p\left(\bar{y}_i \mid x;\ \bar{y}_i \neq \bar{x}_i\right)\right)$$

where bar represents amino acid sequence as before.

## Multihit correction

In the SHM process, having a mutation at a site increases the probability of mutations at nearby sites (*Spisak et al., 2020*). Although a general solution to this problem can easily lead down the path of intractability, we wanted to incorporate this phenomenon into our work.

Inspired by the work of *Lucaci et al., 2021*, we added several simple 'multihit' rate multipliers to our neutral model. These multipliers account for the varying probabilities of mutations based on the number of nucleotide changes required.

We found that, on neutrally evolving out-of-frame data, such a correction substantially improved model fit (*Figure 2—figure supplement 2*). Before correction, the model systematically underestimates multihit mutations. After training the multihit model with correction factors for 1, 2, and 3 mutations per codon, the observed-expected agreement improved substantially for these classes.

This multihit correction was trained on the same separate out-of-frame data as used in training our neutral model (*Sung et al., 2025a*) and was incorporated into the neutral mutation probabilities for DASM training.

## Heavy and light rates

Light chains mutate at a lower rate than heavy chains during affinity maturation. To quantify this relative rate, we used IQ-Tree's rate partition feature during phylogenetic inference on paired heavy-light chain data (see Data section). This allowed us to estimate separate substitution rates for heavy and

light chains within each clonal family. We found the median relative rate of light to heavy chains across all clonal families to be 0.63, which we used as our fixed light chain rate adjustment parameter. This estimate is broadly concordant with previous observations that light chains evolve at roughly half the rate of heavy chains (*Jensen et al., 2024*).

This relative rate is incorporated into the likelihood calculation by scaling the neutral mutation rates for light chain sequences by the light chain rate adjustment factor before combining them with selection factors. Specifically, when evaluating paired heavy-light chain sequences, the per-site neutral mutation rates $p_{j,c}(t, X)$ for light chain sites are multiplied by this adjustment factor, while heavy chain rates remain unscaled.

## Model implementation, training, and evaluation

This model was implemented in PyTorch 2.5 (*Paszke et al., 2019*). Models were trained using the RMSprop optimizer, with 4 cycles, each consisting of branch length optimization followed by neural network optimization.

We used the following software: Altair (*Satyanarayan et al., 2017*; *VanderPlas et al., 2018*), BioPython (*Cock et al., 2009*), Matplotlib (*Hunter, 2007*), pandas (*McKinney, 2010*), pytest (*Krekel et al., 2004*), Seaborn (*Waskom, 2021*), and Snakemake (*Mölder et al., 2021*).

### LLM model score computation

ESM2 scores were calculated using the FLAb paper methodology, with heavy and light chains evaluated separately and their pseudo-perplexities averaged, while AbLang2 directly evaluated paired heavy-light sequences. ProGen2 scores were computed following the same FLAb methodology, in which heavy and light chains are evaluated separately and their perplexities averaged.

### Koenig evaluation

We obtained the Koenig et al. DMS data from the FLAb benchmark repository (*Chungyoun et al., 2024*) (commit 67738ee, April 17, 2024). For DASM evaluation, we used the wild-type G6.31 heavy and light chain sequences as input to predict selection factors for each possible amino acid substitution at each site. The model's log selection factors were compared against the experimental log enrichment ratios via Pearson correlation. We evaluated performance separately for heavy and light chains across both binding and expression datasets.

### Shanehsazzadeh evaluation

We obtained the Shanehsazzadeh et al. binding affinity data from the FLAb benchmark repository (*Chungyoun et al., 2024*) (commit 67738ee, April 17, 2024). We subset to the 'zero-shot' component of the dataset. For DASM evaluation, we partitioned the data by heavy chain amino acid length (119 and 120 residues) due to the diverse nature of the designed sequences. For each length group, we computed a site-by-site consensus sequence to serve as the reference sequence for model predictions. The aggregate log selection factor for each variant was calculated as the sum of log selection factors for each position that differed from the consensus sequence. Model performance was evaluated through Pearson correlation analysis between the aggregate log selection factors and experimental binding affinities.

### MAGMA-seq evaluation

We combined data from two complementary studies using the MAGMA-seq methodology: *Petersen et al., 2024*, providing CDR-targeted mutagenesis data around mature influenza antibodies and *Kirby et al., 2025*, providing UCA to mature antibody evolution trajectories for SARS-CoV-2 antibodies. All sequence variants were assigned to their corresponding antibody systems using reference matching with sequence similarity thresholds. To ensure data quality, experimental replicates were aggregated using geometric mean in $\log_{10}$ space, and high-variance measurements (coefficient of variation > 0.5) were filtered out as in the original papers. After deduplication and quality filtering, the unified dataset contained 1128 sequences across 6 antibody systems: 4 Kirby UCAs and 2 Petersen mature antibodies. The 1_20 antibody system was dropped from the Kirby

data as it only had 7 assigned sequences. Reference sequences were used as inputs for DASM score calculations. Model performance was evaluated using Pearson correlation with binding affinity ($-\log_{10} K_D$).

## Reproducing figures

The following figures were made in notebooks, which can be found in the `notebooks/dasm_paper` directory of the experiments repository.

- *Figure 1a and b* is made in `nt_process_in_llms.ipynb`.
- *Figure 1c*, *Figure 1—figure supplement 1* are made in `koenig_masked_marginals.ipynb`.
- *Table 1* and *Figure 3*, *Figure 3—figure supplements 1–3* are made in `koenig.ipynb`.
- *Figure 3—figure supplement 4* is made by the Snakemake pipeline and in `perplexity.ipynb`.
- *Table 2* and *Figure 3—figure supplement 5* are made in `shanehsazzadeh.ipynb`.
- *Appendix 1—table 1* is made in `data_summaries.ipynb`.
- *Appendix 1—table 2* values are from https://github.com/matsengrp/dasm-experiments/pull/2, copy archived at *Matsen, 2026*.

*Figure 2—figure supplement 2* is made in `multihit_model_exploration.ipynb` in the https://github.com/matsengrp/thrifty-experiments-1, *Sung et al., 2025b* repository.

Our repository file `data/whitehead/MAGMA_PIPELINE_STRUCTURE.md` describes how *Figure 3—figure supplement 6* and *Table 3* are made.

*Table 4* is made by `timing_direct_gpu.py` and `make_timing_table.py`.

## Acknowledgements

We are grateful to the authors of *Tang et al., 2022*, and the lab of Corey Watson for sharing preprocessed data. We thank Antoine Koehl for helpful discussions, Michael Chungyon and Timothy Whitehead for help with data processing, and Aakarsh Vermani for identifying our use of masked-marginals perplexity for the Koenig dataset. This work is supported by NIH grants R01-AI146028 (PI Matsen), R56-HG013117 and R01-HG013117 (PI Song), DP2-AI177890 (PI Starr), and Searle Scholars Program (PI Starr). Scientific Computing Infrastructure at Fred Hutch funded by ORIP grant S10OD028685. Frederick Matsen is an investigator of the Howard Hughes Medical Institute. ChatGPT and Claude AI models were used to draft code and text for this manuscript.

## Additional information

### Funding

| Funder | Grant reference number | Author |
| --- | --- | --- |
| National Institutes of Health | R01-AI146028 | Frederick A Matsen IV<br>Mackenzie M Johnson<br>David H Rich<br>Julia Fukuyama |
| National Institutes of Health | R56-HG013117 | Yun S Song |
| National Institutes of Health | R01-HG013117 | Yun S Song |
| National Institutes of Health | DP2-AI177890 | Tyler N Starr |
| Searle Scholars Program | | Tyler N Starr |
| Howard Hughes Medical Institute | | Frederick A Matsen IV<br>Will Dumm<br>Kevin Sung<br>Hugh K Haddox |

| Funder | Grant reference number | Author |
|---|---|---|
| National Institutes of Health | S10OD028685 | Frederick A Matsen IV<br>Mackenzie M Johnson<br>David H Rich<br>Will Dumm<br>Kevin Sung<br>Hugh K Haddox |

The funders had no role in study design, data collection and interpretation, or the decision to submit the work for publication.

## Author contributions

Frederick A Matsen IV, Conceptualization, Data curation, Software, Formal analysis, Supervision, Funding acquisition, Validation, Investigation, Visualization, Methodology, Writing – original draft, Project administration, Writing – review and editing; Will Dumm, Software, Validation, Investigation, Visualization, Methodology, Writing – review and editing; Kevin Sung, Mackenzie M Johnson, Resources, Data curation, Writing – review and editing; David H Rich, Resources, Software; Tyler N Starr, Yun S Song, Julia Fukuyama, Supervision, Methodology, Writing – review and editing; Hugh K Haddox, Conceptualization, Supervision, Visualization, Methodology, Project administration, Writing – review and editing

## Author ORCIDs

Frederick A Matsen IV, https://orcid.org/0000-0003-0607-6025
Will Dumm https://orcid.org/0000-0002-8617-476X
Kevin Sung https://orcid.org/0000-0002-7289-845X
Mackenzie M Johnson https://orcid.org/0000-0002-3915-2023
Tyler N Starr https://orcid.org/0000-0001-6713-6904
Yun S Song https://orcid.org/0000-0002-0734-9868
Julia Fukuyama https://orcid.org/0000-0002-7590-5563
Hugh K Haddox https://orcid.org/0000-0001-8324-8324

Reviewer #1 (Public review): https://doi.org/10.7554/eLife.109644.3.sa1
Reviewer #2 (Public review): https://doi.org/10.7554/eLife.109644.3.sa2
Reviewer #3 (Public review): https://doi.org/10.7554/eLife.109644.3.sa3
Author response https://doi.org/10.7554/eLife.109644.3.sa4

# Additional files

## Supplementary files

MDAR checklist

## Data availability

Models and inference code can be found at https://github.com/matsengrp/netam; *Matsen et al., 2025a*, including a simple means of accessing the pretrained model demonstrated in the notebooks/dasm_demo.ipynb notebook. Our reproducible experiments are available at https://github.com/matsengrp/dasm-experiments, copy archived at *Matsen, 2026*. Relevant preprocessed data has been uploaded to Zenodo at https://doi.org/10.5281/zenodo.17322891 .

The following dataset was generated:

| Author(s) | Year | Dataset title | Dataset URL | Database and Identifier |
|---|---|---|---|---|
| Matsen IV F, Sung K, Johnson MM | 2025 | Reprocessed data for training and benchmarking Deep Amino acid Selection Models | https://doi.org/10.5281/zenodo.17322891 | Zenodo, 10.5281/zenodo.17322891 |

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

# Appendix 1

## Supplementary data

**Appendix 1—table 1.** Data used in this paper.

CC means that PCPs with mutated cysteines are excluded from the data. `JaffePairedCC` is paired data from *Jaffe et al., 2022*, sequenced using 10×. `TangCC` data is heavy chain data from *Vergani et al., 2017*; *Tang et al., 2022*, and was sequenced using the methods of *Vergani et al., 2017*. `VanwinkleheavyTrainCC1m` is a subset of the heavy chain data from *Engelbrecht et al., 2025*, sequenced using Takara 5'RACE BCR kit. `VanwinklelightTrainCC1m` is a 1M subset of the light chain data from *Engelbrecht et al., 2025*, sequenced using Takara 5'RACE BCR kit. `RodriguezCC` data is the 5' RACE heavy chain data from *Rodriguez et al., 2023*, and is used only for testing. The 'samples' column is the number of individual samples in the dataset; in these datasets, each sample is from a distinct individual. 'Families' is the number of clonal families in the dataset. 'PCPs' is the number of parent-child pairs in the dataset. 'Med. mutns' is the median number of mutations per PCP in the dataset.

| Purpose | Name | Samples | Families | PCPs | Med. mutns |
|---|---|---|---|---|---|
| Train | `JaffePairedCC` | 4 | 50,776 | 209,599 | 7 |
| Train | `TangCC` | 21 | 45,267 | 651,899 | 2 |
| Train | `VanwinkleheavyTrainCC` | 149 | 21,269 | 124,985 | 4 |
| Train | `VanwinklelightTrainCC1m` | 330 | 2658 | 1,000,000 | 2 |
| Test | `RodriguezCC` | 51 | 3592 | 38,050 | 5 |

**Appendix 1—table 2.** Comparison of ESM2 model sizes on the Koenig benchmark using the masked-marginals approach.

The larger ESM2-3B model (2.8B parameters) performs slightly worse than the 650M variant, particularly on light chain predictions.

| Model | Binding | | Expression | |
|---|---|---|---|---|
| | Heavy | Light | Heavy | Light |
| ESM2-650M | −0.001 | 0.308 | 0.418 | 0.524 |
| ESM2-3B | −0.025 | 0.283 | 0.418 | 0.469 |

