## [Editor Report · eLife Assessment]

This **fundamental** study introduces a new biology-informed strategy for deep learning models aiming to predict mutational effects in antibody sequences. It provides **convincing** evidence that separating selection from the nucleotide-level mutation process improves performance over the objectives of protein language models inspired by natural language processing. This paper should be of interest to computational immunologists, but also to the broader community interested in deep learning for biological sequence data and evolution.

---

## [Referee Report · Reviewer #1 (Public review)]

Summary:

Matsen et al. describe an approach for training an antibody language model that explicitly tries to remove effects of "neutral mutation" from the language model training task, e.g. learning the codon table, which they claim results in biased functional predictions. They do so by modeling empirical sequence-derived likelihoods through a combination of a "mutation" model and a "selection" model; the mutation model is a non-neural Thrifty model previously developed by the authors, and the selection model is a small Transformer that is trained via gradient descent. The sequence likelihoods themselves are obtained from analyzing parent-child relationships in natural SHM datasets. The authors validate their method on several standard benchmark datasets and demonstrate its favorable computational cost. They discuss how deep learning models explicitly designed to capture selection and not mutation, trained on parent-child pairs, could potentially apply to other domains such as viral evolution or protein evolution at large.

Overall, we think the idea behind this manuscript is really clever and shows promising empirical results. Two aspects of the study are conceptually interesting: the first is factorizing the training likelihood objective to learn properties that are not explained by simple neutral mutation rules, and the second is training not on self-supervised sequence statistics but on the differences between sequences along an antibody evolutionary trajectory. If this approach generalizes to other domains of life, it could offer a new paradigm for training sequence-to-fitness models that is less biased by phylogeny or other aspects of the underlying mutation process.

Future versions of the work can consider extending the ideas to additional datasets, species, definitions of fitness, or even different proteins entirely.

Comments on revisions:

We thank the authors for addressing our points and have no remaining questions.

---

## [Referee Report · Reviewer #2 (Public review)]

Summary:

Endowing protein language models with an ability to predict the function of antibodies would open a world of translational possibilities. However, antibody language models have yet to achieve the breakthrough success that large language models have achieved for the understanding and generation of natural language. This paper elegantly demonstrates how training objectives imported from natural language applications lead antibody language models astray on function prediction tasks. Training models to predict masked amino acids teaches models to exploit biases of nucleotide-level mutational processes, rather than protein biophysics. Taking the underlying biology of antibody diversification and selection seriously allows disentangling these processes, through what the authors call deep amino acid selection models. These models extend previous work by the authors (Matsen MBE 2025) by providing predictions not only for the selection strength at individual sites, but also for individual amino acids substitutions. This represents a practically important advance.

Strengths:

The paper is based on a deep conceptual insight, the existence of multitude of biological processes that affect antibody maturation trajectories. The figures and writing a very clear, which should help make the broader field aware of this important but sometimes overlooked insight. The paper adds to a growing literature proposing biology-informed tweaks for training protein language models, and should thus be of interest to a wide readership interested in the application of machine learning to protein sequence understanding and design.

Weaknesses:

Proponents of the state-of-the-art protein language models might counter the claims of the paper by appealing to the ability of fine-tuning to deconvolve selection and mutation-related signatures in their high-dimensional representation spaces. Leaving the exercise of assessing this claim entirely to future work somewhat diminishes the heft of the (otherwise good!) argument. In the context of predicting antibody binding affinity, the modeling strategy only allows prediction of mutations that improve affinity on average but not those which improve binding to specific epitopes.

Comments on revisions:

We thank the authors for clarifying the description of the methods and for adding additional discussion of important directions for future work.

---

## [Referee Report · Reviewer #3 (Public review)]

Summary:

This work proposes DASM, a new transformer-based approach to learning the distribution of antibody sequences which outperforms current foundational models at the task of predicting mutation propensities under selected phenotypes, such as protein expression levels and target binding affinity. The key ingredient is the disentanglement, by construction, of selection-induced mutational effects and biases intrinsic to the somatic hypermutation process (which are embedded in a pre-trained model).

Strengths:

The approach is benchmarked on a variety of available datasets and for two different phenotypes (expression and binding affinity). The biologically informed logic for model construction implemented is compelling and the advantage, in terms of mutational effects prediction as well as computational efficiency, is clearly demonstrated via comparisons to state-of-the-art models.

Weaknesses:

While all the main points are well addressed and supported, it could have been interesting to strengthen the claim of gain in interpretability by investigating it explicitly in relation to the functional effects studied in this paper.

Comments on revisions:

I thank the authors for clarifying a few points I had flagged up and I appreciate much better that the content of the companion paper was precisely covering model selection and structural interpretability.

Regarding my first point (references for language models for antibodies), I feel that the parenthetical citation format shouldn't be a problem (but the editors might advise here). Antiberta2 is this paper: https://www.biorxiv.org/content/10.1101/2023.12.12.569610v1.full.pdf (yet, I understand if the authors want to focus on models purely sequence-based). A couple of additional references could be: https://academic.oup.com/bioinformatics/article/40/11/btae659/7888884; https://journals.plos.org/ploscompbiol/article?id=10.1371/journal.pcbi.1012646; https://www.pnas.org/doi/10.1073/pnas.2418918121; https://arxiv.org/abs/2506.13006.

A very minor comment: could one add some p-value (it could be a supplementary table) for the Pearson correlation coefficients? The comparison between methods is rather clear, but for some correlations it's a bit unclear whether they should be considered significant. It would be important to understand the extent to which in different datasets one might expect functional prediction power based on an evolutionary objective function alone.

---

## [Author Response]

The following is the authors’ response to the original reviews

**Public Reviews:**

**Reviewer #1 (Public review):**
Summary:Matsen et al. describe an approach for training an antibody language model that explicitly tries to remove effects of "neutral mutation" from the language model training task, e.g. learning the codon table, which they claim results in biased functional predictions. They do so by modeling empirical sequence-derived likelihoods through a combination of a "mutation" model and a "selection" model; the mutation model is a non-neural Thrifty model previously developed by the authors, and the selection model is a small Transformer that is trained via gradient descent. The sequence likelihoods themselves are obtained from analyzing parent-child relationships in natural SHM datasets. The authors validate their method on several standard benchmark datasets and demonstrate its favorable computational cost.They discuss how deep learning models explicitly designed to capture selection and not mutation, trained on parent-child pairs, could potentially apply to other domains such as viral evolution or protein evolution at large.Strengths:Overall, we think the idea behind this manuscript is really clever and shows promising empirical results. Two aspects of the study are conceptually interesting: the first is factorizing the training likelihood objective to learn properties that are not explained by simple neutral mutation rules, and the second is training not on self-supervised sequence statistics but on the differences between sequences along an antibody evolutionary trajectory. If this approach generalizes to other domains of life, it could offer a new paradigm for training sequence-to-fitness models that is less biased by phylogeny or other aspects of the underlying mutation process.

Thank you for your kind words.

Weaknesses:Some claims made in the paper are weakly or indirectly supported by the data. In particular, the claim that learning the codon table contributes to biased functional effect predictions may be true, but requires more justification.

Thank you for this comment, which made us realize that we had not adequately explained the key insight of Figure S3. We have expanded the caption of Figure S3 to clarify:

“DASM selection factors match the pattern seen in experimental measurements, while masked language models show artifacts from the codon table.

The experimental data (left two panels) show a slight decrease in median scores for amino acids requiring multiple nucleotide mutations (“multiple”) versus single mutations (“single”).

DASM captures this pattern, showing similar distributions for both categories.

In contrast, AbLang and ESM assign radically lower scores to multinucleotide amino acid substitutions, consistent with the masked language modeling objective learning codon-level mutation probabilities as described in the main text (Figure 1a).”

This figure directly supports our claim: the experimental fitness data show similar distributions for single-mutation vs multiple-mutation amino acids, yet AbLang2 and ESM assign dramatically different scores to these groups, while DASM does not.

Additionally, the paper could benefit from additional benchmarking and comparison to enhanced versions of existing methods, such as AbLang plus a multi-hit correction.

It's an interesting idea to consider enhancing existing models. However, this approach faces some challenges. Most fundamentally, it is difficult to recast AbLang and other such models in an evolutionary framework: the masked language objective is simply not an evolutionary one. We have written a whole paper working to do this (https://doi.org/10.1371/journal.pcbi.1013758) and the results were middling despite our best efforts. Specifically regarding multihit, the effects of multihit are minor compared to the codon table effects, and those require the structure of codon-based evolutionary model.

Further descriptions of model components and validation metrics could help make the manuscript more readable.

We have clarified several aspects of the model in the revision: we now describe the Thrifty neutral model in the introduction, clarify the transformer architecture and wiggle activation function in the Methods, and explain the joint branch-length optimization procedure.

In the introduction we now describe Thrifty:

“This fixed model uses convolutions on 3-mer embeddings to deliver wide context sensitivity without needing a large number of parameters: the variant we use has around the same number of parameters as the classic S5F 5-mer model.”

In the Methods we clarify the architecture:

“We parameterize the DASM *f* using the standard transformer-encoder architecture: an amino-acid embedding, sinusoidal positional encodings, and PyTorch's TransformerEncoder module.

The only non-standard component to this architecture is a custom “wiggle” activation function to the output layer that prevents extreme selection factors as previously described.

This function asymptotes to zero for highly deleterious mutations and grows sub-linearly for beneficial ones.”

And the joint optimization:

“This joint optimization is performed cyclically, in which a complete cycle consists of neural network optimization followed by branch length optimization for every parent-child pair.

The parent sequence and the child sequence are pre-estimated, fixed, and used as training data.

The branch lengths are independent and so are optimized in parallel.”

**Reviewer #2 (Public review):**
Summary:Endowing protein language models with the ability to predict the function of antibodies would open a world of translational possibilities. However, antibody language models have yet to achieve breakthrough success, which large language models have achieved for the understanding and generation of natural language. This paper elegantly demonstrates how training objectives imported from natural language applications lead antibody language models astray on function prediction tasks. Training models to predict masked amino acids teaches models to exploit biases of nucleotide-level mutational processes, rather than protein biophysics. Taking the underlying biology of antibody diversification and selection seriously allows for disentangling these processes through what the authors call deep amino acid selection models. These models extend previous work by the authors (Matsen MBE 2025) by providing predictions not only for the selection strength at individual sites, but also for individual amino acid substitutions. This represents a practically important advance.Strengths:The paper is based on a deep conceptual insight, the existence of a multitude of biological processes that affect antibody maturation trajectories. The figures and writing a very clear, which should help make the broader field aware of this important but sometimes overlooked insight. The paper adds to a growing literature proposing biology-informed tweaks for training protein language models, and should thus be of interest to a wide readership interested in the application of machine learning to protein sequence understanding and design.

Thank you for your kind words.

Weaknesses:Proponents of the state-of-the-art protein language models might counter the claims of the paper by appealing to the ability of fine-tuning to deconvolve selection and mutation-related signatures in their high-dimensional representation spaces. Leaving the exercise of assessing this claim entirely to future work somewhat diminishes the heft of the (otherwise good!) argument.

This is an interesting idea! However, it seems to us that this approach has some fundamental limitations. Existing models operate on amino acid sequences with no nucleotide representation, so while they can be implicitly biased by the codon table, they have no signal to separate selection from effects related to the codon table and SHM rates.

We interpret this comment as proposing that we could use fine-tuning on functional data to pull out the selection components (that would only affect the functional data) versus the mutation component. That sounds like an interesting research project. We would be concerned that there are correlations between mutability and selective effects (e.g., CDRs are both more mutable and under different selection), creating identifiability problems unless separate data sources are used as we do here.

Additionally, the fine-tuning approaches we are aware of are taskspecific: they require labeled data from a specific assay (binding to antigen X, expression in system Y) that may or may not relate to the general evolutionary selection signal. Also, such approaches are limited to the specific data used and may not do a good job of guiding the model to a signal that is not present in the training data.

By structuring the model as we do, we obtain the evolutionary interpretation directly from phylogenetic signal without requiring taskspecific supervision.

In the context of predicting antibody binding affinity, the modeling strategy only allows prediction of mutations that improve affinity on average, but not those which improve binding to specific epitopes.

We agree, and this is fundamental to any general purpose model. Predictions of binding patterns for a specific target requires information about that target to be specified in the training data. We look forward to developing such task-specific models in the future.

We have added a paragraph to the Discussion clarifying this limitation:

“The current generation of DASM model does not use any antigen-labeled training data.

The signal that it leverages to infer some limited ability to predict binding comes from natural affinity maturation.

This affinity maturation comes through natural repertoires and so represents a mix of all of the antigens to which the sampled individuals have been exposed.”

**Reviewer #3 (Public review):**
Summary:This work proposes DASM, a new transformer-based approach to learning the distribution of antibody sequences which outperforms current foundational models at the task of predicting mutation propensities under selected phenotypes, such as protein expression levels and target binding affinity. The key ingredient is the disentanglement, by construction, of selection-induced mutational effects and biases intrinsic to the somatic hypermutation process (which are embedded in > a pre-trained model).Strengths:The approach is benchmarked on a variety of available datasets and for two different phenotypes (expression and binding affinity). The biologically informed logic for model construction implemented is compelling, and the advantage, in terms of mutational effects prediction, is clearly demonstrated via comparisons to state-of-the-art models.

Thank you.

Weaknesses:The gain in interpretability is only mentioned but not really elaborated upon or leveraged for gaining insight.

We are also excited about the ability of these models to provide interpretable predictions. We have dedicated an entire paper to this direction: “A Sitewise Model of Natural Selection on Individual Antibodies via a Transformer-Encoder" in MBE (https://doi.org/10.1093/molbev/msaf186). The interpretations offered by that paper overturn some of the oversimplified dogma about how natural selection works in antibodies (purifying in FWK and diversifying in CDR), giving a more nuanced sitewise perspective. The paper also highlights the importance of specific structural features of the antibodies.

This eLife paper, on the other hand, is focused on comparison to antibody language models and benchmarking zero-shot prediction on functional tasks.

We have better highlighted this new paper in our revision with:

“We have dedicated a companion paper to leveraging this interpretability to provide new perspectives on the operating rules of affinity maturation (Matsen et al., MBE 2025): that work provides a nuanced sitewise perspective on natural selection in antibodies that challenges classical oversimplified views of selection patterns.”

The following aspects could have been better documented: the hyperparametric search to establish the optimal model; the predictive performance of baseline approaches, to fully showcase the gain yielded by DASM.

We appreciate the concern and the desire to reveal all the factors that lead to a strong performance result. For this particular paper, we feel that this is less of a concern because we are optimizing according to an evolutionary objective function and then evaluating according to a functional one. We now describe how other than model size, hyperparameters stayed the same as in our previous paper (Matsen et al., MBE 2025).

Regarding baseline approaches, our previous paper includes comparisons to simpler models for the evolutionary objective. Here we focus on comparison to antibody language models for functional prediction. Comparing between state-of-the-art models is the standard practice for papers in this field.

**Recommendations for the authors:**

**Reviewer #1 (Recommendations for the authors):**
We recommend modest amounts of revision, discussed below:Major comments:(1) In the first section of the results, there is extensive discussion on shortcomings of existing antibody language models like AbLang2 that seems to associate all of the performance gap with the inability to separate non-synonymous mutations separated by 1 or 2+ substitutions.In reality, some of the lower likelihoods in the 2+ substitution case could actually reflect real fitness deficits (while others could indeed be rarer occurrences in the training data). The authors should either moderate these claims or do an analysis that leverages antibody deep mutational scanning data to show that, conditioned on the fitness of the antibody (probably expression) being the same (either all high or all low), AbLang2 still artefactually considers rarer-training/less-codon-accessible variants to be less fit.

As described above, we believe that this is addressed by Figure S3, but if not please correct us.

(2) Some in the machine learning for antibody community might view the set of benchmarked datasets to be incomplete and somewhat arbitrarily selected, though we do think this is a good start, and the results are promising. A dataset commonly used in this field that is missing from this paper is from Shehata et al. (https://pubmed.ncbi.nlm.nih.gov/31553901/). A binding affinity experiment that is also commonly used in the field is from Phillips et al. (https://elifesciences.org/articles/71393) - this dataset measures combinatorial changes of framework regions on binding, which may be especially relevant here.

We're glad to have the opportunity to clarify this, thanks.

We based our evaluations on the April 2024 version of the FLAb benchmarking project (https://doi.org/10.1101/2024.01.13.575504) which preceded our work and thus was not subject to selection bias by us. We took the largest data sets in that repository. After this we became aware of the rich data sets offered by the Whitehead lab that provided binding measurements for many variants for a number of antigens, and added that to the evaluation set.

We have clarified this in the manuscript:

“We based our evaluations on the April 2024 version of the FLAb benchmarking project, which preceded our work and thus was not subject to selection bias by us.

We also benchmarked high-throughput binding data (more recent than FLAb) from the Whitehead lab that provided affinity measurements across many variants and antigens.”

The Shehata dataset is interesting but doesn't fit so much in the DASM mold: it is a survey of biophysical properties across many independent antibodies rather than a deep investigation of point mutants of a smaller collection of focal antibodies.

FLAb has grown to include the Phillips dataset. We are working full-tilt on the next version of DASM and will be including many other datasets in our paper on DASM2. Thanks for the tip!

(3) Similar to the above comment, we were also extremely curious as to why the authors did not test data from DeWitt et al. (https://pubmed.ncbi.nlm.nih.gov/40661619/). Instead, the authors only make a cryptic reference to this study on lines 201-6, but we could not even find a figure describing the results discussed on these lines. It would be great to actually include this data.

We agree, however, our model is for human rather than mouse. We would like to train a mouse model in the future but have not yet lined up the appropriate data.

(4) The authors should comment on potential data leakage if the SHM trajectories used in training have a similar sequence or antigen similarity to the benchmark expression/binding datasets.

This is a good question that we should clarify. Our model is trained only on evolutionary trajectories and not functional data. Evaluation is then done on functional data without fine-tuning. Because these evaluation data are categorically different from the training data and thus data leakage is not a problem. Recall that our model is zero-shot: it only considers evolutionary trajectories and not functional data as such. In a similar way, other self-supervised models such as MLMs do not exclude seeing an antibody in the training data when they are doing functional prediction.

We have clarified this in the manuscript with

“Because the DASM is trained exclusively on evolutionary trajectories rather than functional measurements, evaluation on expression and binding benchmarks is strictly zero-shot with no risk of data leakage.”

Relatedly, what happens if this approach is applied to completely de novo antibodies?

We direct this reviewer to the Shanehsazzadeh dataset that involves antibodies that were suggested by an AI algorithm rather than observed in nature.

If the reviewer is referring to completely synthetic antibody molecules, such as those generated by inverse folding, we have not attempted this.

(5) It makes sense that you included the multihit correction as a response to your earlier instantiation (without this correction) underestimating the probabilities of multiple mutations in a codon associated with a single amino acid substitution (lines 476-477).However, this could potentially make for a somewhat unfair comparison to existing methods: if, say, we took AbLang (or another comparator) and also applied a multi-hit correction (even in some naive way at inference time), how would that compare to DASM? If this comparison favors DASM, it would show that models need more than just such a correction on top of existing methods to do good sequence scoring--which would only amplify the impact of the results.

Thank you for this suggestion. We believe that we have addressed it in the response to the public reviews, but please let us know if not.

Minor comments:(1) It would be worth explicitly defining/summarizing the mutation model used in the study, e.g. giving an overview of Thrifty in the introduction or where it first appears.

Thanks, we have done this:

“Our approach separates mutation and selection processes by encoding functional effects in a Deep Amino acid Selection Model (DASM) while explicitly modeling mutation using a separate fixed model trained on neutrally evolving data.

This fixed model uses convolutions on 3-mer embeddings to deliver wide context sensitivity without needing a large number of parameters: the variant we use has around the same number of parameters as the classic S5F (Yaari et al., 2013) 5-mer model.”

(2) Paragraph starting on line 58: it sounds like you're suggesting that masked deep learning models will learn certain features of genomes in a certain order. We suggest that you weaken the language, giving examples of various things the model could learn, not implying that such models will necessarily learn the most useful features after the less useful ones.

We have fixed this by removing the "First... Second... Third... Finally" ordering:

“It could memorize the germline genes and learn about the probabilities of V(D)J recombination.

It could learn the codon table, as according to this table some aminoacid mutations are much more likely than others. It could learn rates of somatic hypermutation...

It could also learn about the impact of amino acid mutations on antibody function through natural selection in the course of affinity maturation, which is the desired signal.

However, this desired signal is confounded by the preceding factors.”

(3) Line 72: You make a strong claim that existing models conflate mutation and selection without knowing for sure that they didn't successfully learn these components separately (it seems this would require a lot of mechanistic interpretability). The language could be softened here.

We believe that we have addressed this in the response to public reviews, but please let us know if not.

(4) Line 79: Say a bit more about the separate fixed mutation model here. Why shouldn't we worry about this choice (especially the word "fixed") biasing your results? Does the empirical performance of your method suggest this doesn't really matter?

We have added to the description of the fixed mutation model, as described above.

As described in the public response, training SHM models on out-of-frame sequences is an established methodology for characterizing mutation in the absence of selection. In principle one could jointly train a model of SHM and selection, but one could have identifiability problems as there is a correlation between more mutable sites (e.g. in the CDRs) and those under relaxed selection. Using out-of-frame sequences gives a clean an independent description of the SHM process.

(5) Line 81: on what benchmarks does it outperform? State briefly.

Great suggestion. Done:

“The DASM, trained on substantially less data, outperforms AbLang2 and general protein language models including ESM2 and ProGen2-small. This outperformance holds on the largest benchmark datasets of the FLAb collection and on recent high-throughput binding assays.”

(6) Paragraph starting on line 90: The topic sentence reads a bit vague to us. Do you mean that you want to learn the extent to which models are regurgitating nucleotide similarity of AAs in determining the scores associated with AAs at masked sites?

Thank you. We have updated to

"We first sought to understand the extent to which processes such as neutral mutation rate and the codon table influence antibody language model prediction at masked sites."

(7) Paragraph starting on line 108: feels speculative and maybe better for the discussion...

We appreciate this comment, but we have decided to keep the content where it is. Although this would make sense as a Discussion item we feel like it fits well here right next to the evidence, and the structure of our Discussion doesn't really have a place for it.

(8) Paragraph starting on line 116: don't say "sequences from [12]" or "method of [15]." Explain what these are before giving the citation.

Whoops! Thanks. We have fixed these.

(9) Line 134: Consider giving a brief definition of perplexity?

Thanks. We added our favorite definition:

“Perplexity (as defined in the Methods) is the standard way of evaluating the plausibility of a sequence according to a model: it is the acrosssite geometric mean of the inverse probability of the observed amino acid.”

(10) Line 154: A citation here could be useful to support the claim that these models are learning phylogeny.

We have replaced with the more clearly established "codon table":

“We implemented a model to learn amino-acid preferences of antibodies without being influenced by germline genes, the codon table, or SHM biases.”

(11) Lines 161-162: Given that phylogenetic inference methods can be tough to scale, we're curious how you managed to get 2 million PCPs from the data? Did you construct a bunch of different phylogenies (in > parallel)?

Indeed! We now clarify in the methods section that these trees were run in parallel across clonal families:

“As in our previous work, tree inference and ancestral sequence reconstruction were performed per clonal family with the K80 substitution model...

Because these clonal families are independent these phylogenetic inferences were run in parallel.”

(12) Line 173-174: Can you say more about the joint optimization of the branch lengths? Are you conditioning on a phylogenetic tree topology only, and leaving the branch lengths unknown? Do you account for the fact that these branch lengths in the same phylogenetic tree aren't independent?

Thanks for pointing out the need to clarify these points. We have done so in the methods section and provided a pointer to the methods section in the main text.

In the main text we now say:

“We trained DASMs of several sizes (~1M, ~4M, ~7M) using joint optimization of branch length *t* and parameters of the DASM (see Methods for details).”

And in the Methods:

“This joint optimization is performed cyclically, in which a complete cycle consists of neural network optimization followed by branch length optimization for every parent-child pair.

The parent sequence and the child sequence are pre-estimated, fixed, and used as training data.

The branch lengths are independent and so are optimized in parallel.”

(13) Line 358: Yes, in a trivial sense, separating mutation and selection means that we know exactly how each of those two components has been learned. We would be curious if you could say anything about mechanistic interpretability within the deep learning selection model. If not, could this be a future research direction?

We believe that we have addressed this in the response to public reviews, but please let us know if not.

(14) Lines 384-386--indeed. Do you have any proposals for how a phylogeny could be constructed at this scale?

As above this is not one big phylogeny but many, which invites parallelization.

**Reviewer #2 (Recommendations for the authors):**
(1) I agree that a full study of fine-tuning strategies for all possible alternative models is beyond the scope of the paper. However, a little bit of fine-tuning would go a long way to demonstrate how easy (or hard) it is to extract the relevant signal from a general protein language model embedding.

As described in our response to the public reviews, we appreciate this point but have decided to focus on the core novelty of the paper and leave fine-tuning experiments to future work.

(2) The authors might want to add some discussion about what signals their models capture with regard to binding affinity (averages), and how this limitation might be addressed in future work.

As described in our response to the public reviews, we have added a paragraph to the Discussion clarifying this limitation.

**Reviewer #3 (Recommendations for the authors):**
(1) Introduction: I think more references have to be provided re: Antibody "foundation" language models, e.g. adding AntiBERTy and the two versions of AntiBERTa.

We have added citations to those two models, although we weren't sure what the second version of AntiBERTa was. There are very many antibody language models. If we could use number ranges we would cite a dozen or more, but I hesitate to add many of them in the eLife format, which has parenthetical citations. If there are others that you consider essential don't hesitate to suggest them.

(2) A key point of the approach is the disentanglement of “mutation” and “selection”, as mentioned in the introduction. However, the explanation of what the authors mean by mutation and selection comes only later. I would anticipate it in the introduction for clarity.

This is a great point. The revised intro has this in the second sentence:

“Natural antibodies are generated through V(D)J recombination, and refined by somatic hypermutation and affinity-based selection in germinal centers.”

and the "While the masked..." paragraph now more clearly calls out selection.

(3) Line 133: expression of what? Could the authors also explain mechanistically why expression should be impacted by a mutation? In what conditions do these data sample expression?

We have clarified that it is expression in a phage display library:

“To do so, we used the largest dataset of the FLAb collection of benchmarks, which measures the effect of single mutations on expression in a phage display library.”

(4) Line 142: Clarify that 0.49 and 0.3 are correlation coefficients. Also, what type of correlation coefficient is this?

Thanks for the catch! They are Pearson correlations as we now describe.

(5) Line 173: The hyperparametric search should have been more documented (with a description of how it was carried out and plots).

As described in our response to the public reviews, we are optimizing according to an evolutionary objective function and then evaluating according to a functional one. Other than model size, hyperparameters stayed the same as in our previous paper (Matsen et al., MBE 2025).

(6) Line 358: The authors say that 'DASMs provide direct interpretability'. However, this is not really inspected. A valuable addition would be to show how such interpretability is made possible, how it can recapitulate existing biological knowledge or provide hints for antibody engineering.

As described above, this is addressed in detail in our previous paper.

(7) Line 398: 'Inferred insertions or deletions were reversed, so that all sequences align to the naive sequence without gaps.' Could the authors comment on whether this is a limitation of the approach, why it wasn't dealt with and whether it could be the direction of future work?

Funny you should mention this! We have been planning out such an extension in detail recently. We have added a sentence in the discussion:

“We also have plans to extend the DASM framework to estimate the effect of natural selection on insertion and deletion events.”

(8) Line 430-431: Could the authors clarify 'shared' over what? Also, I believe these two lines really describe the DASM architecture. This should be spelt out more clearly and tied to the description provided in lines 173-175. A diagram of the architecture would be a valuable addition to provide a full picture of the model (this could be added to the general diagram of the modelling approach of Figure S8).

We have clarified in the text that this is indeed a description of the DASM architecture -- thanks for the catch:

“We parameterize the DASM *f* using the standard transformer-encoder architecture: an amino-acid embedding, sinusoidal positional encodings, and PyTorch's TransformerEncoder module.

The only non-standard component to this architecture is a custom “wiggle” activation function to the output layer that prevents extreme selection factors as previously described.”

The architecture is very “stock” - just the default torch TransformerEncoder, so I don't think that it merits a diagram. We have expanded our discussion of the simple architecture in the revision. This sits in contrast to the setup for the loss function, which is quite custom and is the subject of Figure 2 and Figure S8.

(9) Another general remark is that, to fully showcase the predictive advantage offered by DAMS with all the modelling choices entailed, one could show the performance of simpler models, like the mutation model alone (with no selection factors), or models where selection factors are just learnt independently for each site, or are learnt with a simple linear layer instead of a transformer (these are just ideas of some simpler approach that can set baselines over which DASM improvement can be shown).

This is a great suggestion. The primary focus of this paper is in comparing to alternate antibody language models in terms of functional prediction.

These simpler models could be used for comparing the evolutionary objective, which we did in our previous paper (https://doi.org/10.1093/molbev/msaf186). We note that a sitewise model with fixed sites cannot really be appropriately formulated due to sequences being of different lengths.

Additional changes

In addition to the reviewer-requested changes, we added a comparison of ESM2 model sizes (650M vs 3B parameters) on the Koenig benchmark. We found that scaling ESM2 from 650M to 3B parameters did not improve performance. Indeed, the larger model showed slightly degraded correlations, particularly for light chain predictions. This is consistent with recent observations that medium-sized protein language models can outperform larger ones on transfer learning tasks (Vieira et al., Sci. Rep. 2025). We added Table S2 documenting these results and cite this finding in the main text to justify our use of the 650M model throughout the analyses. After doing this, we realized for the Shanehsazzadeh evaluation we had accidentally used ESM2-3B instead of ESM2-650M. The corrected ESM2-650M values are slightly lower (0.191 and 0.308 for sequence lengths 119 and 120, respectively, compared to the previous values of 0.248 and 0.337). This correction does not affect our conclusions, as DASM substantially outperforms ESM2 on this benchmark before and after the change.

We also realized in the course of revision that we had been scoring AbLang2 using the masked-marginals pseudo-perplexity approach for the single-mutant Koenig dataset (Figure 1c), rather than the standard persequence pseudo-perplexity used elsewhere in the paper. For maskedmarginals, probabilities are computed using only wild-type context, whereas standard pseudo-perplexity uses each variant's own context.

The masked-marginals approach has a simple interpretation: for singlemutation variants, it is a linear transformation of the log ratio of the variant amino acid probability to the wild-type amino acid probability, both evaluated under wild-type context. This log-odds ratio directly measures how much the model prefers the mutation over the original residue.

We found that masked-marginals performed better for AbLang2 on this dataset, so we continued using it for Figure 1c. However, for the benchmarking table (Table 1), we switched to per-sequence pseudoperplexity as for the other comparisons in the paper, following the standard benchmarking protocol defined in FLAb (Chungyoun et al., 2024). We document both approaches in the Methods section:

“An alternative “masked-marginals” approach scores variants using only wild-type context.

For a wild-type sequence *w*, masked-marginals computes \begin{document}$p\left(a \mid w_{\backslash i}\right)$\end{document}. for all amino acids *a* at each position *i* once, then uses these wild-type-derived probabilities to compute pseudoperplexity for any variant *x*...

For a single-mutation variant *x* that differs from wild-type *w* only at position *j*, all terms except position *j* cancel when comparing to wild-type, giving \begin{document}$(x))]$\end{document}. Thus, the log-probability difference between variant and wild-type amino acids equals, up to an additive constant \begin{document}$n \backslash \log (\backslash t e x t\{m m-p s e u d o-p p l\}(w))$\end{document} that depends only on the wild-type sequence, negative *n* times the log pseudo-perplexity of the variant.

For Figure 1c on the single-mutant Koenig dataset, we found that this approach gave a higher correlation for AbLang2 and so used it in that figure.

For benchmarking comparisons (Table 1), we followed standard practice and used per-sequence pseudo-perplexity.”